# A Triband Slot Patch Antenna for Conformal and Wearable Applications

**Erfeng Li** *, **Xue Jun Li** * and **Boon-Chong Seet**

Department of Electrical and Electronic Engineering, Auckland University of Technology, Auckland 1010, New Zealand; boon-chong.seet@aut.ac.nz

\* Correspondence: brandt.li@aut.ac.nz (E.L.); xuejun.li@aut.ac.nz (X.J.L.)

**Abstract:** With the rapid development of wireless communication technology and the Internet of Things (IoT), wireless body area networks (WBAN) have been thriving. This paper presents a triband patch antenna with multiple slots for conformal and wearable applications. The proposed antenna operates at 5.8, 6.2, and 8.4 GHz. The antenna was designed with a flexible polyethylene terephthalate (PET) substrate, and the corresponding conformal tests and on-body performance were conducted via simulation. The antenna demonstrated promising gain and acceptable fluctuations when applied on curvature surfaces. The specific absorption rate (SAR) for on-body simulation also suggests that this antenna is suitable for wearable applications.

**Keywords:** triband; flexible; slot antenna; patch antenna; Internet of Things; wearable; wireless body area network; specific absorption rate





## 1. Introduction

With the recent rapid development, the latest wireless communication technology can provide a high data rate and ubiquitous connectivity [1], which provides unlimited opportunities for the internet of things (IoT) [2–4]. Consequently, wireless body area networks (WBAN) have been thriving [5–7], and the demand for wireless wearable devices is increasing in physical training, healthcare monitoring [8,9], military tracking, etc. [10,11]. Thus, as a key component of a wireless communication system, an antenna plays an irreplaceable role. For WBAN applications, the antenna needs to have a compact and flexible structure and a wide operating frequency range.

Microstrip slot patch antennas have been a popular research topic in both industry and academia ever since their first appearance [12–14]. With great features, such as a compact size, low-cost fabrication, and light weight, microstrip slot patch antennas have become widely applied in printed circuit board (PCB) designs [15,16]. In [17], a microstrip patch Yagi antenna was reported at 5 GHz. It employed slotted patches as a parasitic patch reflector, and short pins were also applied in the design. The overall size of the proposed antenna and the soldering pins made it difficult to be flexible.

A circularly polarized slot patch antenna was presented in [14] for the frequency band from 3.2 to 3.6 GHz. The antenna consisted of a circular ground plane and a slotted square patch and was fed by coaxial feeding. The antenna could achieve high gain, but the bulky structure was not suitable for compact and flexible applications. In [18], a hexaband circularly polarized slotted patch was proposed that operated at six frequency bands for wireless local area network (WLAN), long term evolution (LTE), and radio navigation applications. The antenna employed an S-shaped slot on the patch to achieve multiple operating frequency bands. However, the two layers of rigid FR4 substrates is not suitable for conformal and wearable applications.

After a careful literature review, we noticed that a compact flexible antenna can be used in WBAN applications with curvature structures, such as wearable healthcare monitoring devices. Motivated by the advantages of microstrip slot patch antennas, we propose a

flexible slot patch antenna with tuning slots, which enables the antenna to resonate at three frequency bands above 5 GHz. Furthermore, simulations where the antenna is applied on curvature surfaces and human body are conducted and discussed.

For IoT applications, the efficacy and efficiency of Machine-to-Machine (M2M) devices are important. The communication interest and interaction among different M2M devices should be considered to form coalitions [19]. The multiple frequency band feature of the proposed antenna guarantees that the device with this antenna can communicate with other devices in the IoT system seamlessly without interference.

The rest of this paper is organised as follows: Section 2 presents the major design principle of the patch antenna, followed by Section 3, which elaborates on the parametric study of the design. Next, Section 4 discusses the antenna performance. After that, a further evaluation simulation for conformal application is provided in Section 5, followed by the on-body performance for wearable applications in Section 6. Last, Section 7 concludes this paper.

## 2. Proposed Patch Antenna Design

A triband slot microstrip patch antenna is proposed with conductive ink-printing for conformal applications. The antenna has three operating bands, namely 5.8, 6.2, and 8.4 GHz respectively. The substrate material chosen for this design is a flexible PET film, which has potential for conformal applications. The triband slot antenna is based on a microstrip patch structure, consisting of radiating patch on the top and ground plane at the bottom, isolated by a PET dielectric layer (dielectric constant $\varepsilon_r = 3.66$, thickness $h = 1$ mm) in the middle. The patch dimensions are determined via Equations (1)–(3) with respect to the lowest frequency of 5.8 GHz [20]:

$$\varepsilon_{eff} = \frac{\varepsilon_r + 1}{2} + \frac{\varepsilon_r + 1}{2}\left[1 + 12\frac{H}{W}\right]^{-1/2}, \tag{1}$$

$$L_{eff} = L + 2\Delta L, \tag{2}$$

$$\frac{\Delta L}{h} = 0.412\frac{(\varepsilon_{reff} + 0.3)\left[\dfrac{W}{h} + 0.264\right]}{(\varepsilon_{reff} - 0.258)\left[\dfrac{W}{h} + 0.8\right]}, \tag{3}$$

where $W$ and $L$ are the patch width and length, $h$ is the substrate thickness and $\varepsilon_r$ is the dielectric constant of the substrate material. The antenna is fed by a 50-$\Omega$ microstrip line.

## 3. Parametric Study for Multiband Slots

Adding slots to the patch will significantly affect the surface current distribution, thus, altering and even adding more resonant frequencies [15,16,20–25]. The theoretical mechanism of a slot on a patch is similar to that of the equivalent dipole antenna, where the slot is regarded as the reciprocal dipole. Based on the above fundamental concept, a parametric study of the slots on the patch was performed via simulation.

First, a ring slot is etched on the patch as shown in Figure 1. The width of the slot ring $W_{slot}$ is set to 0.504 mm based on the aforementioned dipole theory, and the space between the ring and the patch edge $S_{slot}$ is set as 1.02 mm.

Secondly, to achieve multiple bands, jointing parts are needed to separate the slot ring into different parts. Two major parametric simulation are conducted to achieve this step by step. Taking the top left corner of the ring slot as a reference point, the distance between joint1 and the reference point $Y_{joint1}$ and the length of joint1 $L_{joint1}$ are investigated, where parametric sweeps are conducted in the range of 0–10 mm and 0–5 mm, respectively.

The corresponding resonant frequency response and return loss is given in Figures 2 and 3. Making the position of joint1 fixed by setting $Y_{joint1}$ as 0, 2, 4, and 6 mm, parametric sweeps are conducted with different values of $L_{joint1}$.

Clearly there is a resonant frequency band near 5 GHz for all the four cases in Figure 2. Another frequency band near 6 GHz appears stable in the second sweep as shown in Figure 2, whereas the third frequency band near 8 GHz is flickering and ambiguous across the four sweeps. In Figure 2, only one case completely shows three frequency bands at $L_{joint1} = 5$ mm and $Y_{joint1} = 0$ mm. To summarise the impact of joint1 length $L_{joint1}$, it mainly affects the 8 GHz frequency band.

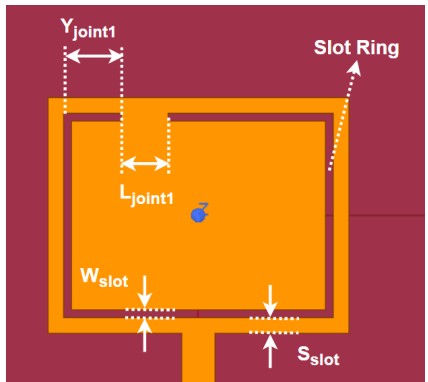

**Figure 1.** Adding the first joint part on the top of the ring slot.

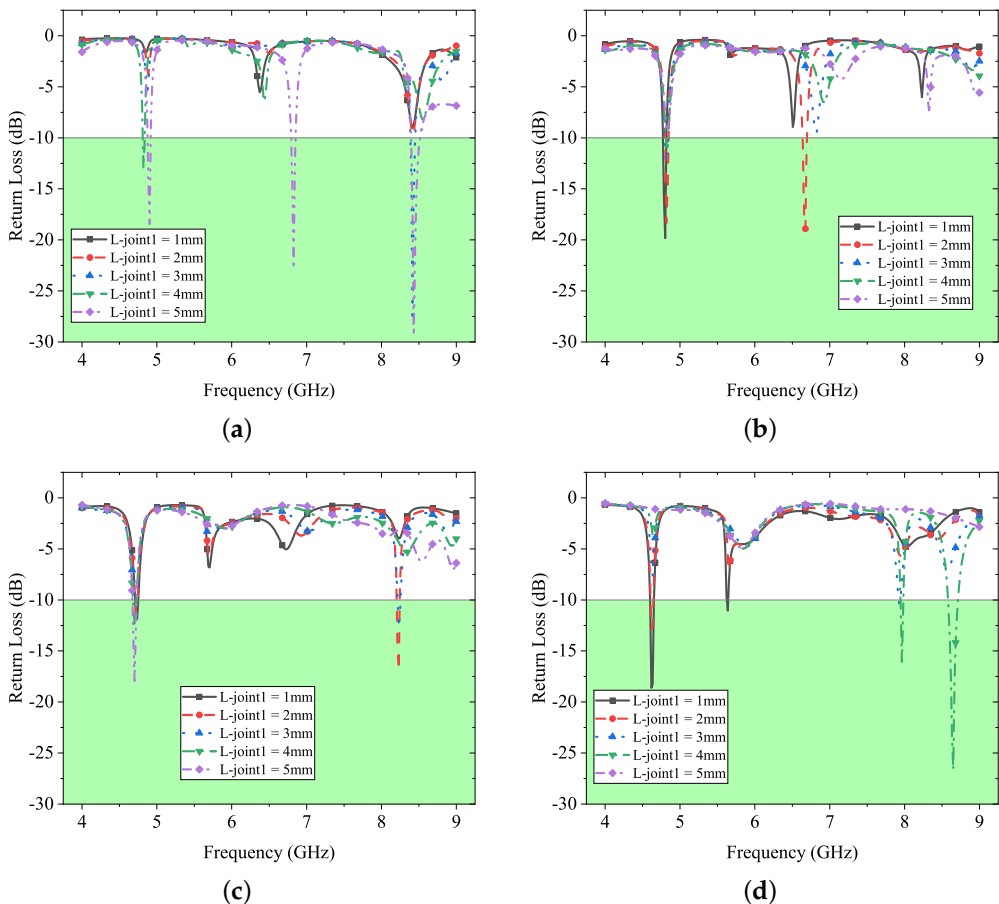

**Figure 2.** Effect of joint1 length ($L_{joint1}$) on return loss performance. (**a**) Parametric sweep when $Y_{joint1} = 0$ mm; (**b**) Parametric sweep when $Y_{joint1} = 2$ mm; (**c**) Parametric sweep when $Y_{joint1} = 4$ mm; and (**d**) Parametric sweep when $Y_{joint1} = 6$ mm.

The parametric sweeps of the joint1 position along the *y*-axis are shown in Figure 3, by setting the joint length $L_{joint1}$ as 1, 2, 3, and 4 mm, respectively. The near 5 GHz band is relatively stable in each sweep scenario. Additionally, a band near 8 GHz also shows in most of the sweep cases, and only a few cases fulfil the −10 dB standard. Furthermore, the resonance of the 6 GHz band is not strong enough.

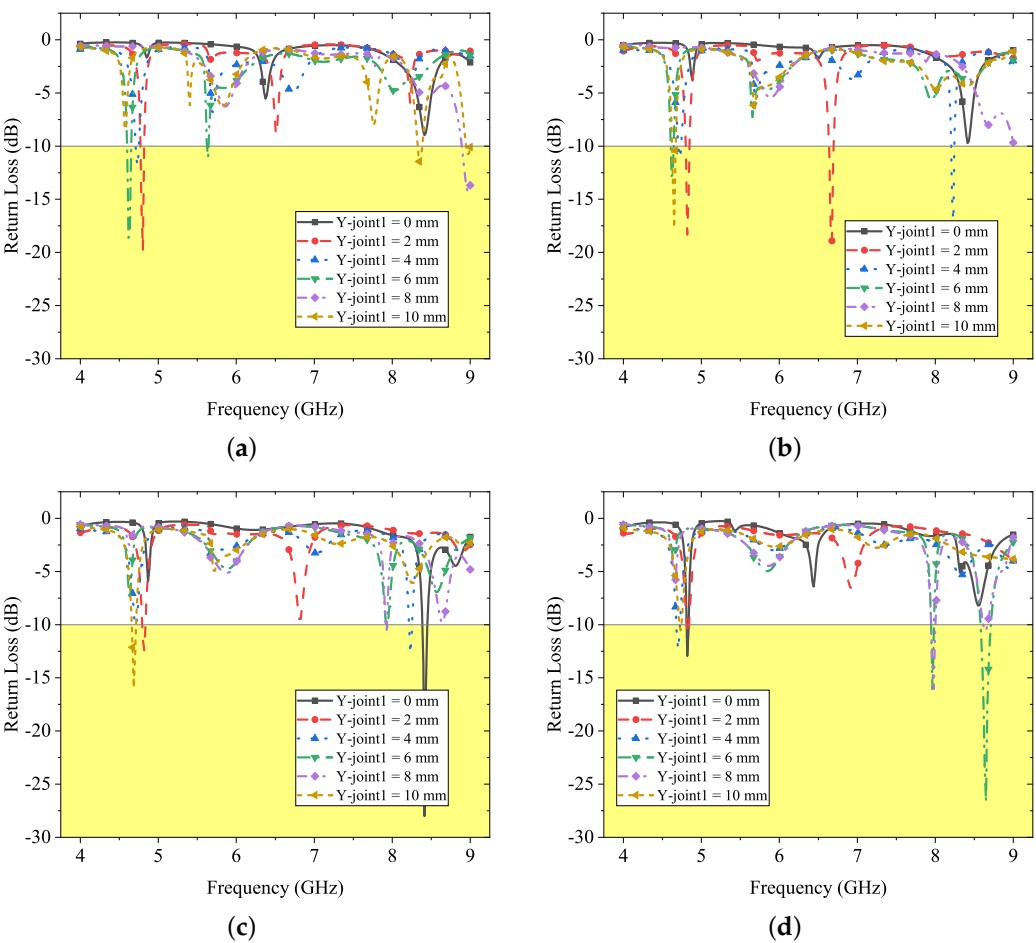

**Figure 3.** Effect of the joint1 position ($Y_{joint1}$) on the return loss performance. (**a**) Parametric sweep when $L_{joint1}$ = 1 mm; (**b**) Parametric sweep when $L_{joint1}$ = 2 mm; (**c**) Parametric sweep when $L_{joint1}$ = 3 mm; and (**d**) Parametric sweep when $L_{joint1}$ = 4 mm.

The above results regarding $L_{joint1}$ and $Y_{joint1}$ can also be validated in the parametric sensitivity report in Figure 4, which presents the magnitude of the partial derivative of $|S_{11}|$ with respect to these two parameters, respectively. As indicated in the highlighted areas, in Figure 4, the magnitude of reflection coefficient $|S_{11}|$ is very sensitive to $L_{joint1}$ at the frequency band of 8.1–8.4 GHz; in Figure 4, $|S_{11}|$ reacts to $Y_{joint1}$ much more at the band of 4.5–5 GHz and 8–8.4 GHz. This agrees with the above parametric sweeps of those two parameters. Consequently, one can conclude that adding joint1 yields a frequency band at near 8 GHz, which may affect the 6 GHz band as well.

Altogether, these above results suggest that adding joint1 gives a second frequency band at near 8 GHz, which is sensitive to the *y*-axis location $Y_{joint1}$ and length $L_{joint1}$ of the joint. These two parameters will be fine tuned later together with those of joint2.

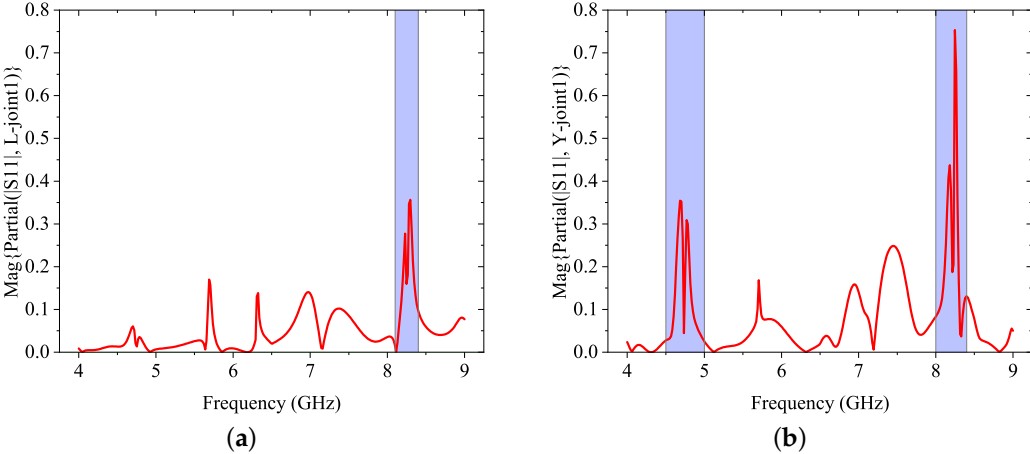

**Figure 4.** Parameter sensitivity of $Y_{joint1}$ and $L_{joint1}$. (**a**) Magnitude of the partial derivative of $S_{11}$ over $L_{joint1}$; and (**b**) Magnitude of the partial derivative of $S_{11}$ over $Y_{joint1}$.

Thirdly, a second joint is added on the right of the ring slot as shown in Figure 5. Considering the bottom right corner of the ring slot as a reference point, the distance between joint2 and the reference point $X_{joint2}$ and the length of joint2 $L_{joint2}$ are also swept in HFSS simulation in the range of 0–3 mm and 0–8 mm, respectively.

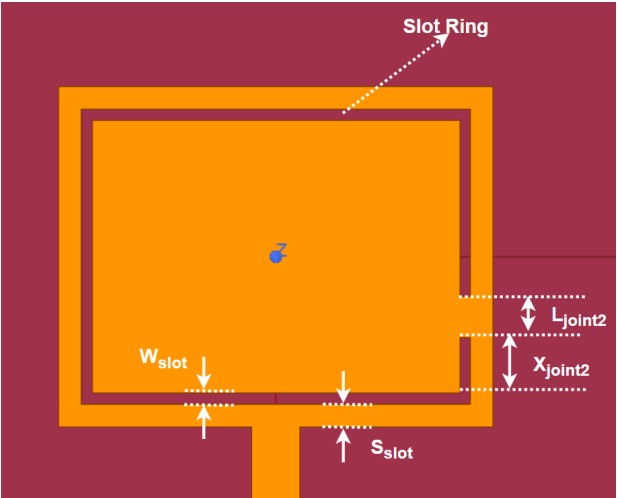

**Figure 5.** Adding second joint part at the bottom right of ring slot.

Figure 6 presents the influence on the antenna's resonant frequencies from $L_{joint2}$. The results of four parametric sweeps are presented in the diagram, while $X_{joint2}$ is set as 0, 1, 2, and 3 mm, respectively. In the first three scenarios in Figure 6a–c, three resonant frequencies can be found at near 6, 7, and 8 GHz, respectively. However, the band near 7 GHz is flickering and not quite sensitive. Additionally, the band near 8 GHz in Figure 6 and the band near 6 GHz in Figure 6 are relatively stable. For the scenario in Figure 6, the band near 8 GHz is relatively stable, and no resonant frequency can be found apart from that. These observations suggest a link between $L_{joint2}$ and a resonant frequency band near 7 GHz.

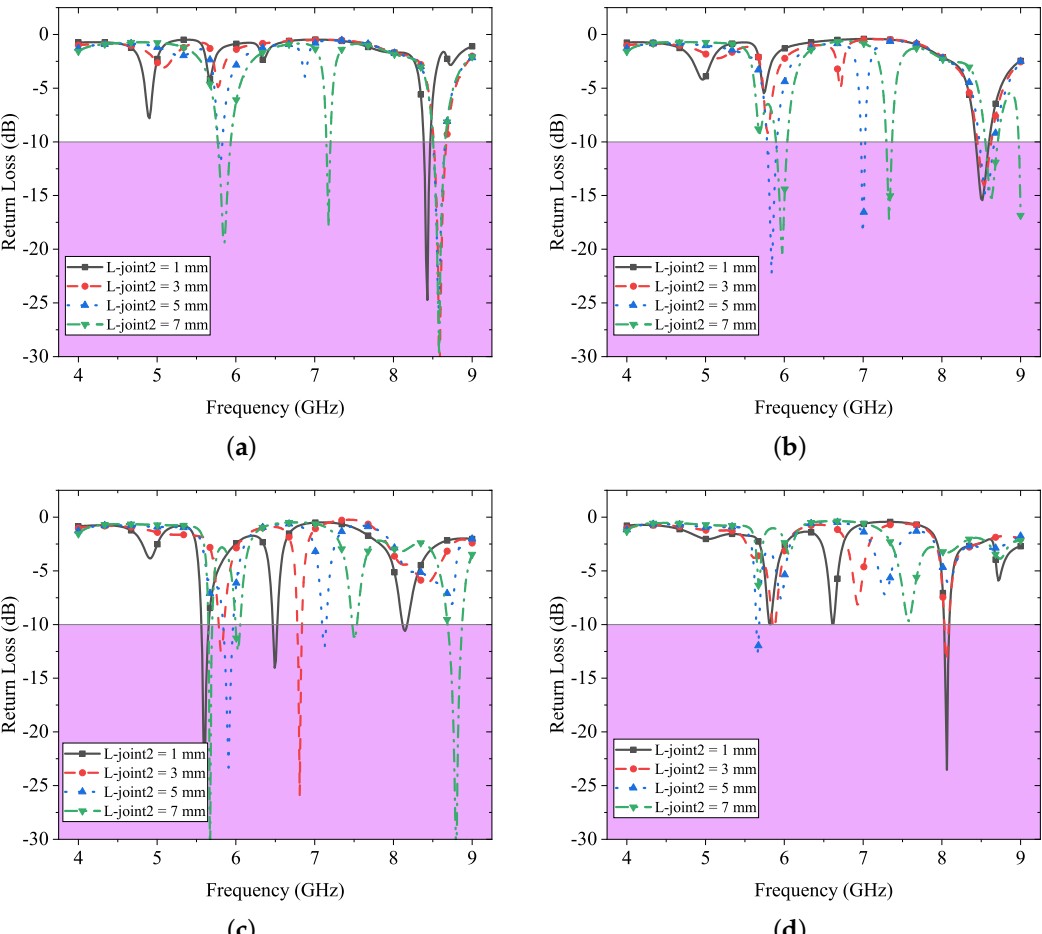

**Figure 6.** Effect of the joint2 length ($L_{joint2}$) on return loss performance. (**a**) Parametric sweep when $X_{joint2} = 0$ mm; (**b**) Parametric sweep when $X_{joint2} = 1$ mm; (**c**) Parametric sweep when $X_{joint2} = 2$ mm; and (**d**) Parametric sweep when $X_{joint2} = 3$ mm.

Next, in Figure 7, the joint2 length $L_{joint2}$ is set fixed as 1, 3, 5, and 7 mm, respectively for the parametric sweep of $X_{joint2}$. In the four scenarios, one can observe three resonant frequency bands approximately at near 6, 7, and 8 GHz, especially in Figure 7c,d, the three bands are distinguishable, while those in the other two cases are not steady and not matching the −10 dB criteria, especially the near 7 GHz band, which is floating between 6–7 GHz. It shows that $X_{joint2}$ is another important factor for the middle band near 6 and 7 GHz.

From Figure 8, one can observe the parametric sensitivity of those two parameters of joint2. In Figure 8a, it shows that the magnitude of the reflection coefficient $|S_{11}|$ is very sensitive to $L_{joint2}$ at frequency band of 6.5–7 GHz as shown in the highlighted area; In Figure 8b, the impact from $X_{joint2}$ on $|S_{11}|$ at near 6 GHz is at about the same level of that from $L_{joint2}$. These above observations are in agreement with the previous parametric sweeps. However, the impact at 7.9–8.2 GHz band is surprisingly much higher, which may suggest that $X_{joint2}$ will not only affect the 6–7 GHz band but also the near 8 GHz band.

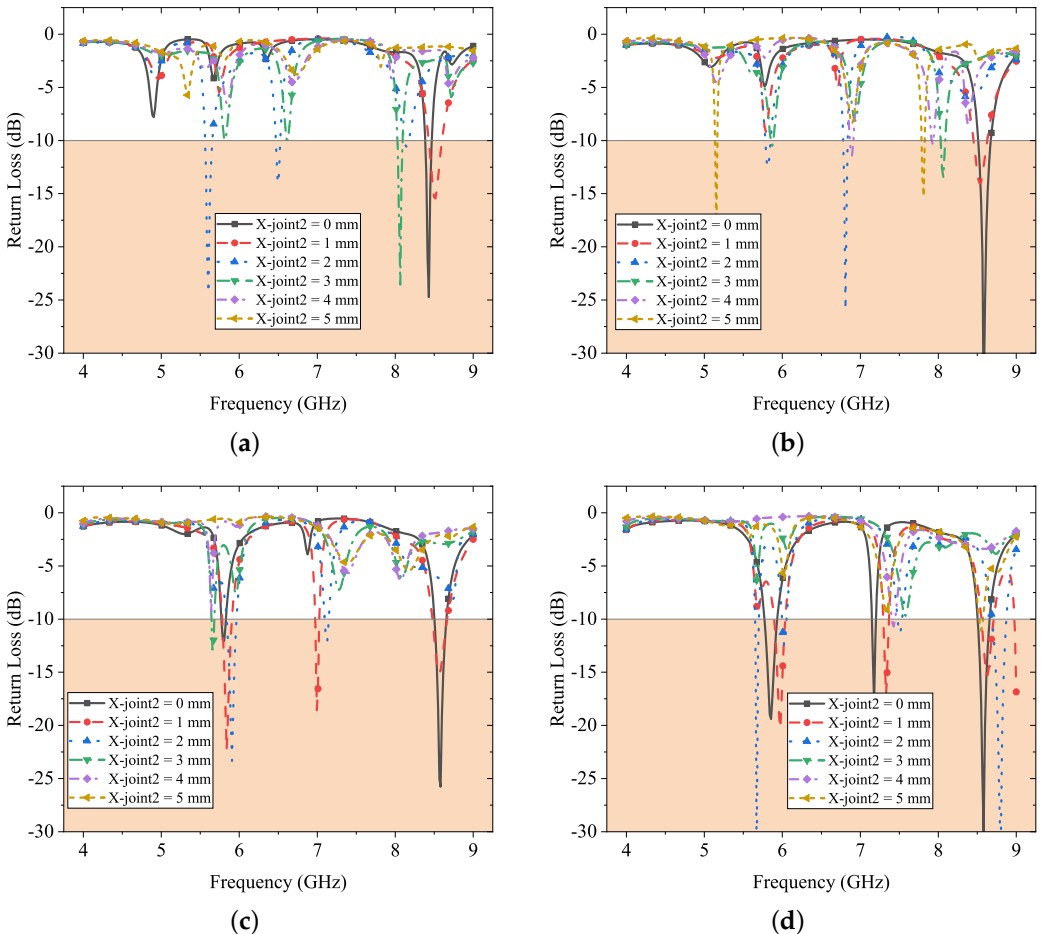

**Figure 7.** Effect of the joint2 position ($X_{joint2}$) on return loss performance. (**a**) Parametric sweep when $L_{joint2}$ = 1 mm; (**b**) Parametric sweep when $L_{joint2}$ = 3 mm; (**c**) Parametric sweep when $L_{joint2}$ = 5 mm; and (**d**) Parametric sweep when $L_{joint2}$ = 7 mm.

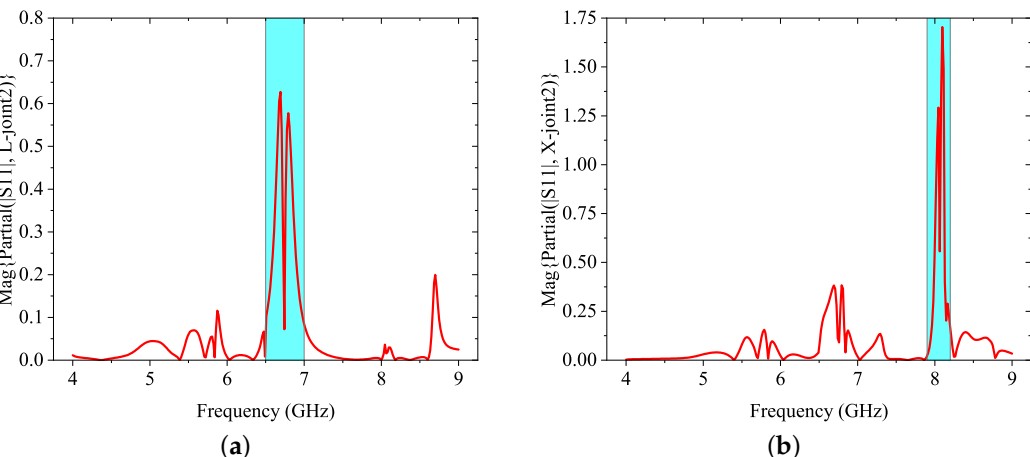

**Figure 8.** Parameter sensitivity of $X_{joint2}$ and $L_{joint2}$. (**a**) Magnitude of the partial derivative of $S_{11}$ over $L_{joint2}$; and (**b**) Magnitude of the partial derivative of $S_{11}$ over $X_{joint2}$.

By optimising the parameters of the two joints, the antenna dimensions are finalised as: $L_p$ = 15.37 mm, $W_p$ = 19.64 mm, $L_f$ = 12.32 mm, $W_f$ = 2.15 mm, $t_1$ = 0.504 mm, $t_2$ = 8.192 mm, $t_3$ = 3.282 mm, $t_4$ = 3.808 mm, and $t_5$ = 10.74 mm. The overview of the finalised antenna is shown in Figure 9a, and it operates at three resonant frequency bands at 5.8, 6.2, and 8.4 GHz.

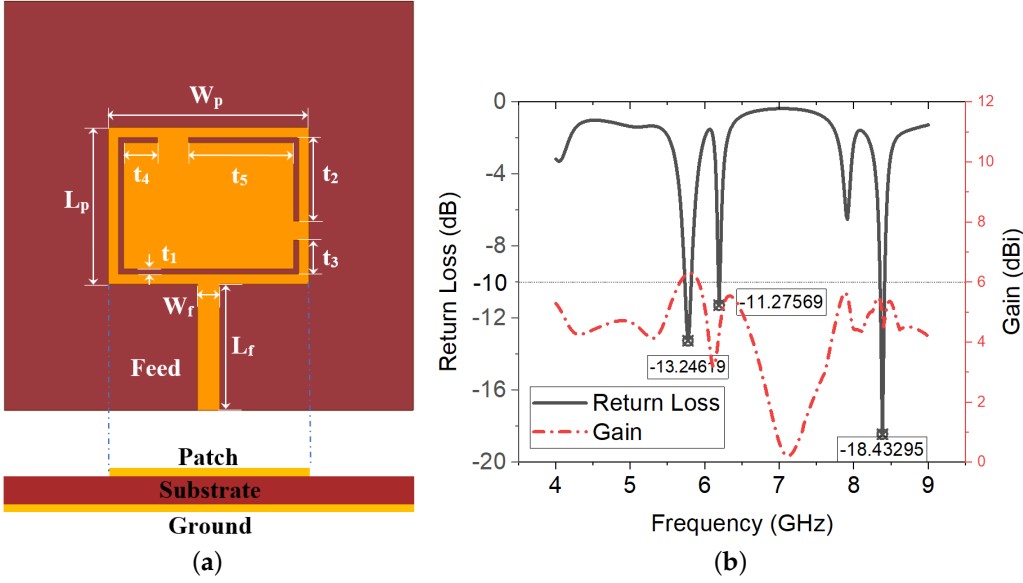

(a)            (b)

**Figure 9.** Finalised antenna layout and antenna performance. (**a**) Triband slot antenna structure layout. (**b**) Simulated return loss and gain of the proposed antenna.

## 4. Results and Discussion

The proposed antenna is evaluated in ANSYS HFSS. The return loss and antenna gain illustrated in Figure 9b show that the antenna operates at three frequency bands ($|S_{11}| < -10$ dB), i.e., 5.8, 6.2, and 8.4 GHz, respectively; bandwidth values are 91, 10, and 65 MHz, respectively; the corresponding antenna gain at these operating frequencies are 6.23, 4.62, and 5.43 dBi, respectively.

Next, the patch surface current distribution at 90° phase and the radiation patterns of E-plane and H-plane at these operating frequencies are also illustrated in Figure 10a to Figure 10f, respectively. From the patch surface current distribution, it can be seen that the current with maximum magnitude is predominantly scattered around the slots, where the resonance is happening correspondingly.

At 6.2 GHz, there is a relatively large area in the centre of the patch with low current distribution, suggesting the poor resonance and narrow frequency bandwidth indicated in Figure 9b. As for the radiation patterns, due to the asymmetry of the slots along feed line direction, the E-plane (phi = 0°) patterns are symmetrical but not the H-plane (phi = 90°) patterns.

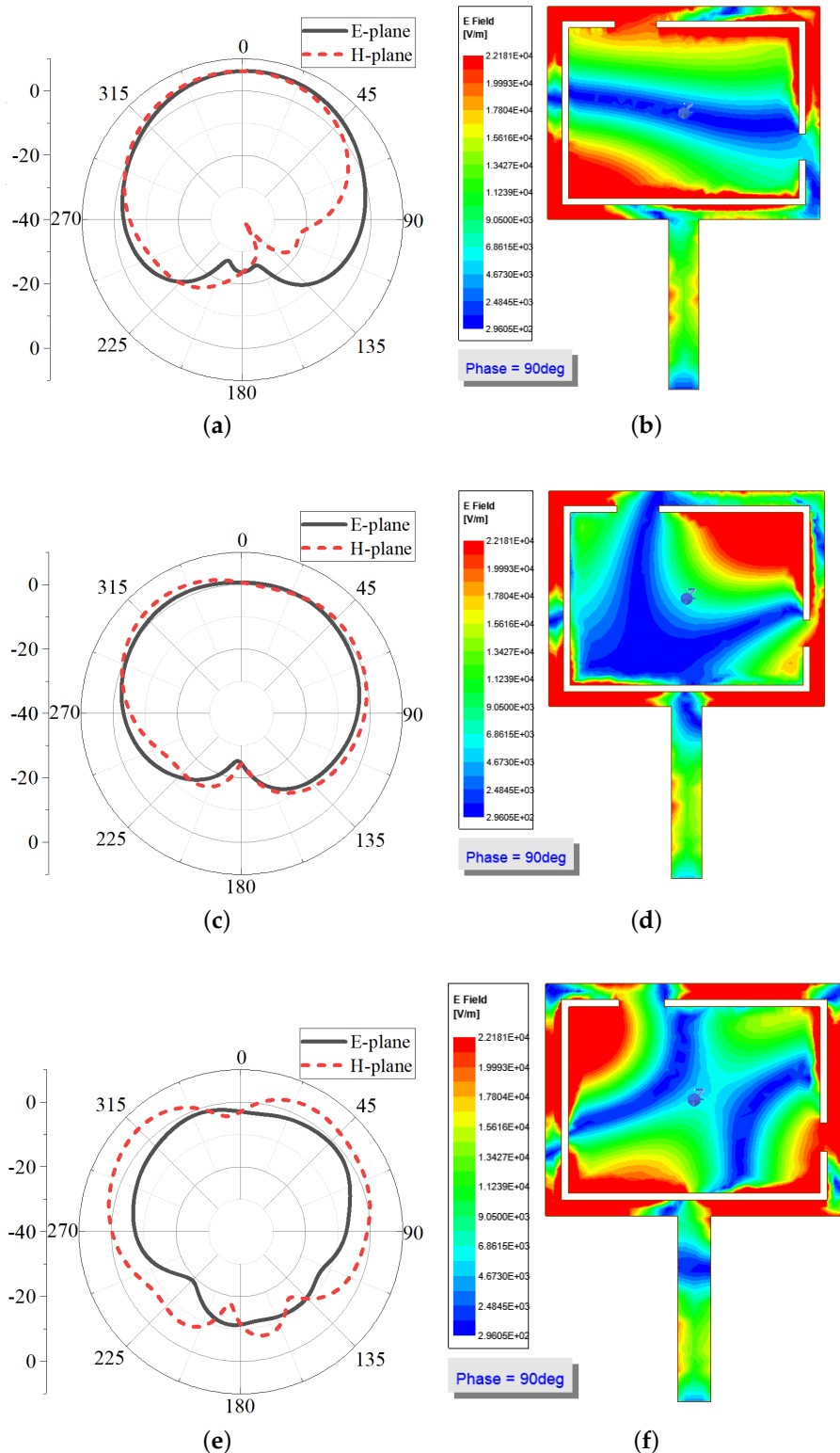

**Figure 10.** Simulated antenna performance at three bands. (**a**) Simulated radiation pattern at 5.8 GHz; (**b**) Surface current distribution at 5.8 GHz; (**c**) Simulated radiation pattern at 6.2 GHz; (**d**) Surface current distribution at 6.2 GHz; (**e**) Simulated radiation pattern at 8.4 GHz; and (**f**) Surface current distribution at 8.4 GHz.

## 5. Conformal Application Test

To evaluate the performance of the antenna under flexible surface circumstance, a set of conformal simulations are conducted, where the entire antenna is wrapped onto a cylinder along $y$-axis. To simulate different levels of surface bending, the radius of the cylinder is set as discrete values at 20, 50, and 100 mm. The corresponding bending angles can be approximated as 18°, 7.2°, and 4.5°, respectively. The cylinder is set as a non-model object, which has no effects on Maxwell's equation calculations or the simulation results.

The frequency response of the antenna and return loss are illustrated in Figure 11 for the three bending tests. There is a clear resonant frequency at 5.8 and 6.2 GHz, and the return loss values are below −15 dB. However, the resonance at 8.4 GHz is weak when the bending cylinder radius is 20 mm, which means that most of the energy is reflected by the antenna. The antenna appears to be affected significantly by large angle bending.

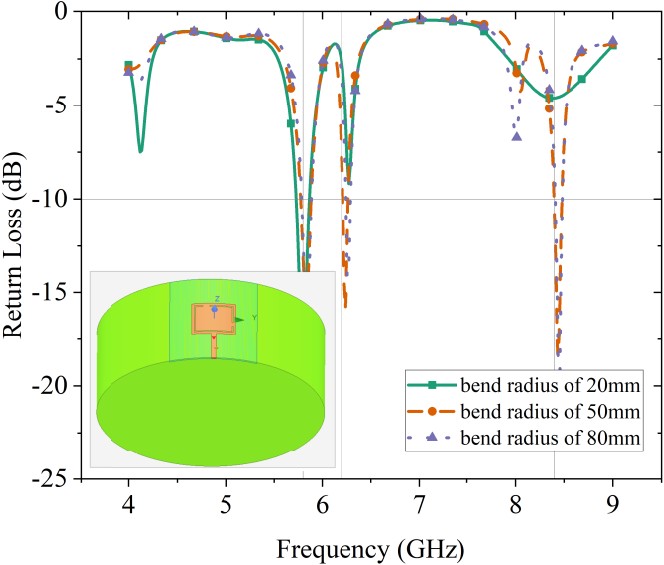

**Figure 11.** Simulated antenna return loss for different bending radii.

When the bending angle is 18° (or cylinder radius is 20 mm), the 3D polar plots of E-field of the antenna at three frequency bands are illustrated in Figure 12a–c. It can be seen that the radiation pattern at 5.8 GHz still has a major lobe concentrating and pointing in the +$z$-axis direction, whereas the one for both 6.2 GHz splits into one major lobe pointing in the −$y$-axis direction and a minor side lobe pointing in the +$y$-axis direction.

As for the plot of 8.4 GHz in Figure 12c, the lobe splits into two approximately equal sized major lobes, pointing at −$y$-axis and +$y$-axis, respectively. This suggests that the directivity and antenna gain at 6.2 and 8.4 GHz are altered by the bending effects. This is due to the fact that the bending force folds the antenna into two parts around the $x$-axis; therefore, the radiated field also tends to diverge along the bending.

Moreover, the maximum E-field strength for each frequency band is 14.4, 8.6, and 12.2 V/m, respectively, which indicates the radiation intensity of each case. It is obvious that, at 5.8 GHz, the antenna radiates the strongest electromagnetic field, followed by 8.4 and then 6.2 GHz. This matches the corresponding return loss at each resonant frequency mentioned earlier.

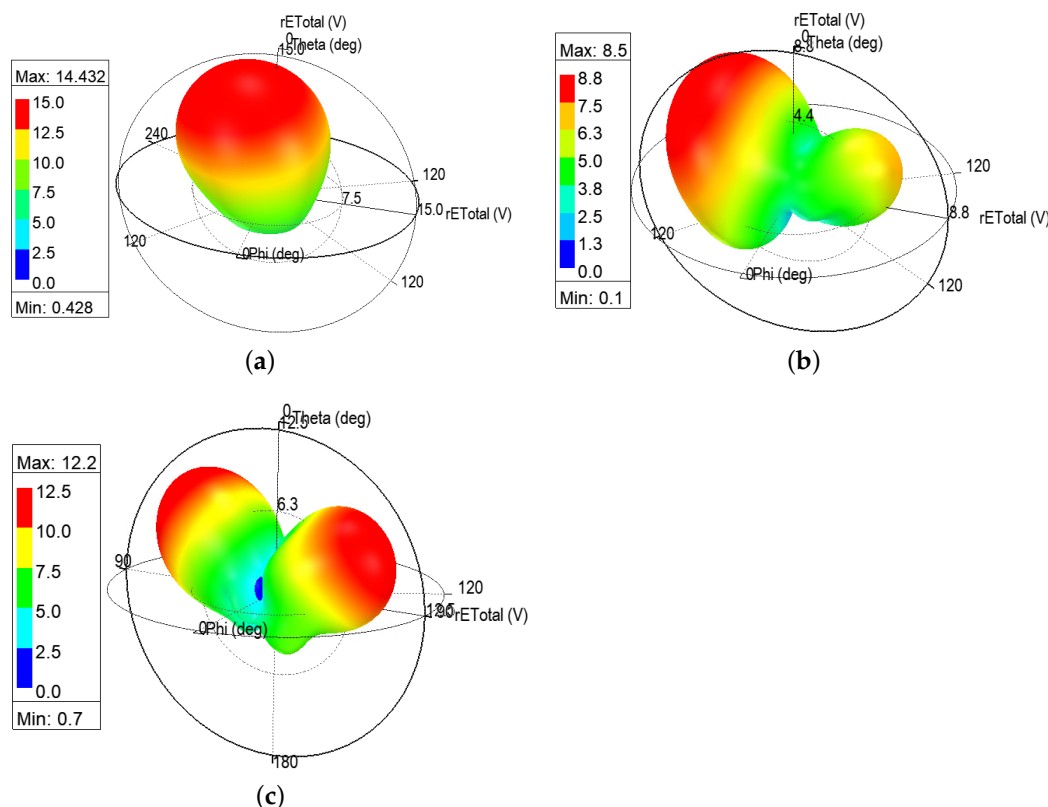

**Figure 12.** Electric field 3D polar plot for cylinder radius of 20 mm. (**a**) Electric field 3D polar plot at 5.8 GHz; (**b**) Electric field 3D polar plot at 6.2 GHz; and (**c**) Electric field 3D polar plot at 8.4 GHz.

Regarding the case of cylinder radius 50 mm and bending angle 7.2°, the 3D polar plots of E-field are demonstrated in Figure 13a–c. The general patterns of E-field 3D polar plot at 5.8 and 8.4 GHz remain similar to those of previous case where cylinder radius is 20 mm. A significant pattern change happens at 6.2 GHz, where most of the E-field concentrates above the radiating patch, pointing along the +*z*-axis, but it also shows a tendency to split into two lobes. The maximum E-field strengths were measured as 15.2, 10.6, and 14.6 V/m, respectively. At the 6.2 GHz band, the antenna radiates relatively weaker electromagnetic fields.

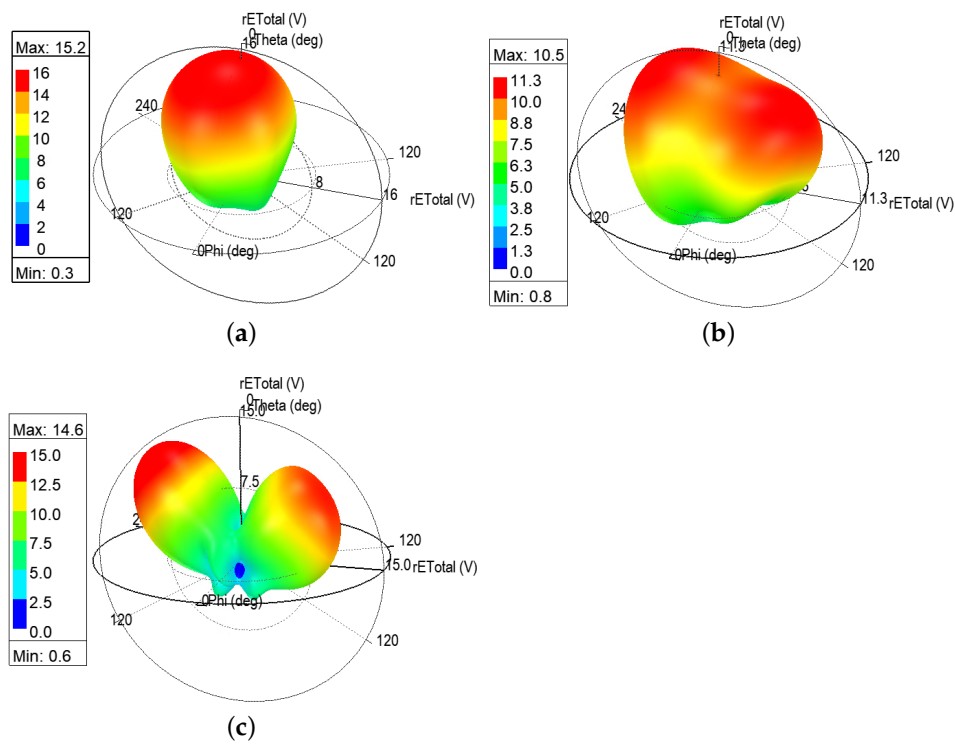

**Figure 13.** Electric field 3D polar plot for cylinder radius of 50 mm. (**a**) 3D polar plot of electric field at 5.8 GHz; (**b**) 3D polar plot of electric field at 6.2 GHz; and (**c**) 3D polar plot of electric field at 8.4 GHz.

When it comes to the case of the 80 mm cylinder radius, there are not significant pattern changes in the E-field 3D plots at 5.8 and 8.4 GHz compared with the case of the 50 mm cylinder radius. Interestingly, Figure 14b reveals a gradual deviation of the E-field strength pattern, which is the tendency of main lobe splitting into two is getting higher. It approaches a similar shape to that at 8.4 GHz. Overall, the maximum E-field strengths are 15.2, 8.4, and 13.9 V/m. The radiation performance at 5.8 and 8.4 GHz is still stronger than that at 6.2 GHz.

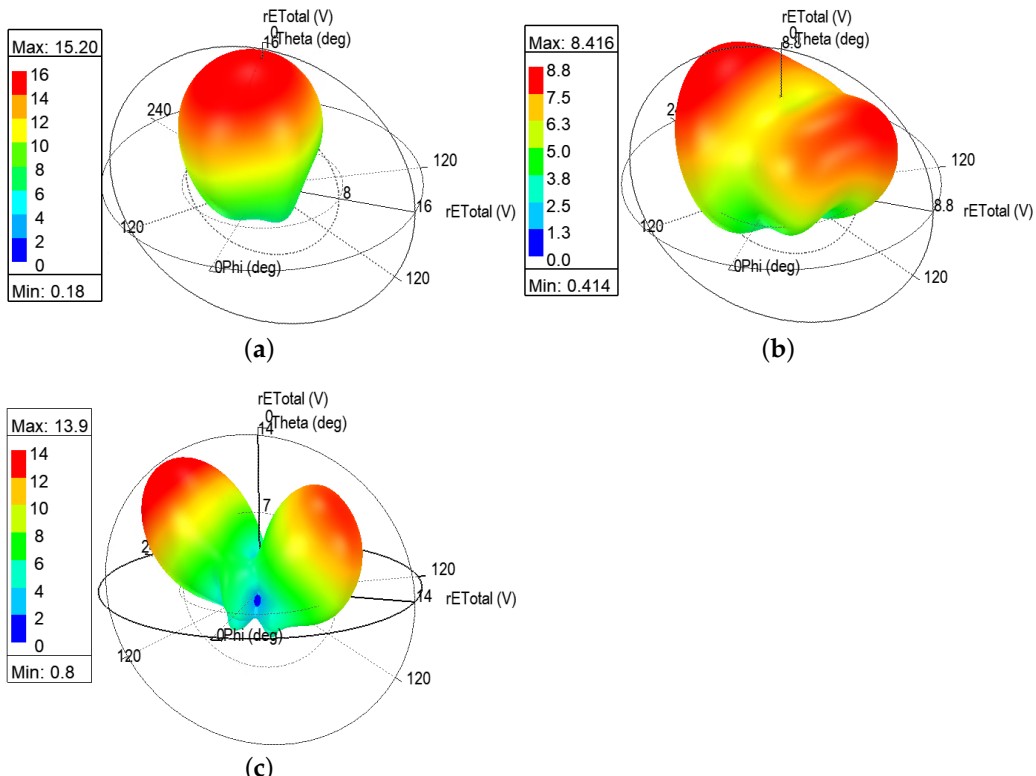

**Figure 14.** Electric field 3D polar plot for cylinder radius of 80 mm. (**a**) Electric field 3D polar plot at 5.8 GHz; (**b**) Electric field 3D polar plot at 6.2 GHz; and (**c**) Electric field 3D polar plot at 8.4 GHz.

After a careful study of the principle E-plane and H-plane radiation patterns in Figure 15, one can conclude that only minor changes can be found in the radiation patterns at 5.8 GHz. For the case of 6.2 GHz, the overall pattern of E-plane increases, whereas that of H-plane shrinks as the cylinder radius increases, which suggests the E-field radiation has noticeable improvement. Furthermore, the size of back lobe of the E-plane drops to a lower level. As for the frequency band of 8.4 GHz, the E-plane radiation patterns changed significantly when the bending angle is large for the cylinder radius of 20 mm. At the same time, the directivity is obvious for the 8.4 GHz band. The H-plane patterns have two predominantly major lobes and two small back lobes as well.

As further observation, the surface current distribution on the radiating patch is presented for the bending case of cylinder radius of 50 mm. Figures 16–18 represent the cases for three frequency bands at 5.8, 6.2, and 8.4 GHz, respectively. Different phases of the power source are listed in each sub-figure, i.e., 0°, 30°, 60°, 90°, 120°, and 150°, respectively.

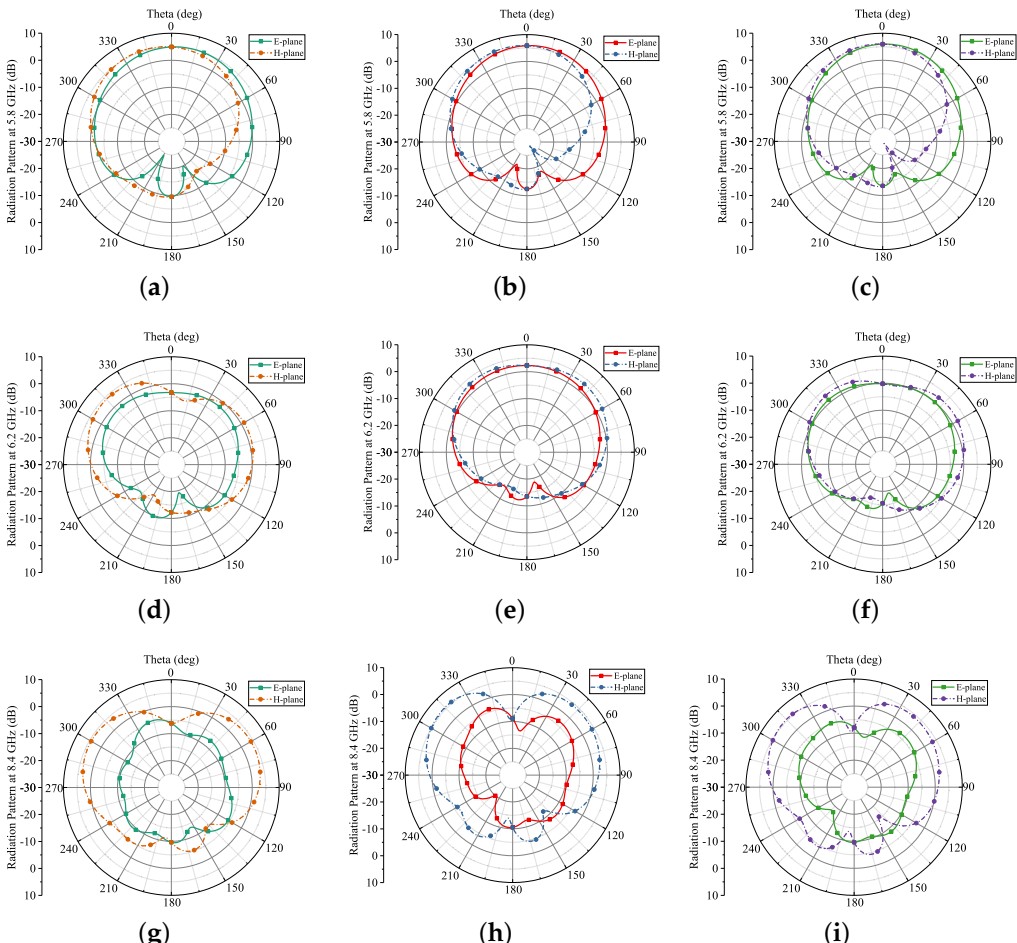

**Figure 15.** Simulated E-plane and H-plane for conformal tests. (**a**) E, H-plane at 5.8 GHz for 20 mm; (**b**) E- and H-plane at 5.8 GHz for 50 mm; (**c**) E- and H-plane at 5.8 GHz for 80 mm; (**d**) E- and H-plane at 6.2 GHz for 20 mm; (**e**) E- and H-plane at 6.2 GHz for 50 mm; (**f**) E- and H-plane at 6.2 GHz for 80 mm; (**g**) E- and H-plane at 8.4 GHz for 20 mm; (**h**) E- and H-plane at 8.4 GHz for 50 mm; and (**i**) E- and H-plane at 8.4 GHz for 80 mm.

At 5.8 GHz, when the phase is 90° and 120°, the overall current distribution is low and below 26 A/m. For other phases, the surface current concentrates around the edge of the patch and vicinity of the slot, up to above 200 A/m. At 6.2 GHz, the current distribution at different phases is similar to that of 5.8 GHz but with higher overall concentration and coverage.

Deviations can be found in Figure 18 regarding the surface current distribution at 8.4 GHz. The current level at 60° and 90° is lower than that of other phases, whereas the current level at 0°, 30°, and 150° is much more intense and higher than in the other two frequency bands.

Overall, these results regarding the surface current distribution suggest that, under the effect of both frequency and the bending effect, the current flow inside the radiating patch varies, and the low current area contributes to the conduction loss of the antenna. The high level of time-varying current on the patch at 8.4 GHz also corresponds to the relatively high level induced magnetic field shown in the radiation patterns in Figure 15.

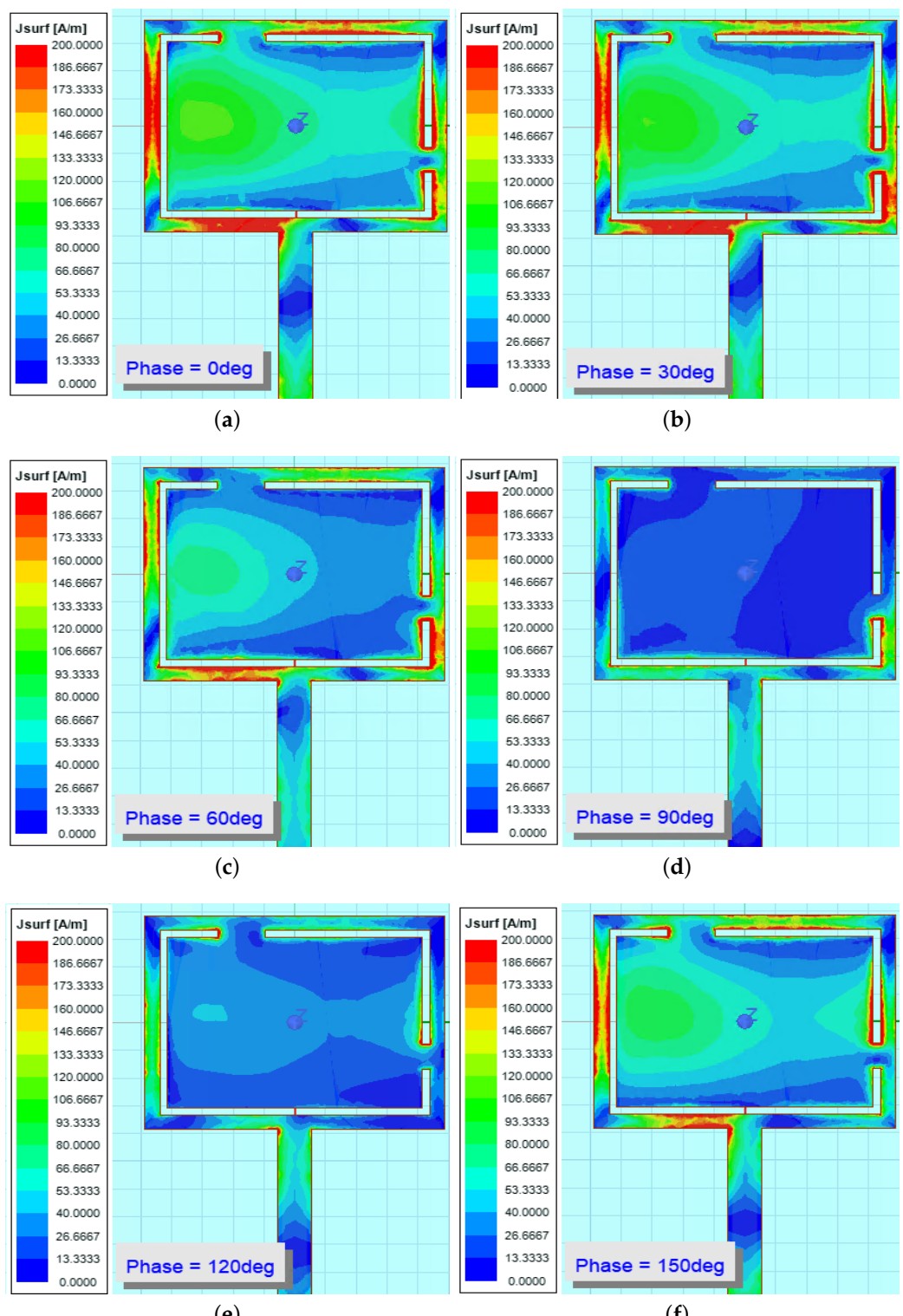

**Figure 16.** Surface current distribution at 5.8 GHz for cylinder radius of 50 mm. (**a**) 0 degrees; (**b**) 30 degrees; (**c**) 60 degrees; (**d**) 90 degrees; (**e**) 120 degrees; and (**f**) 150 degrees.

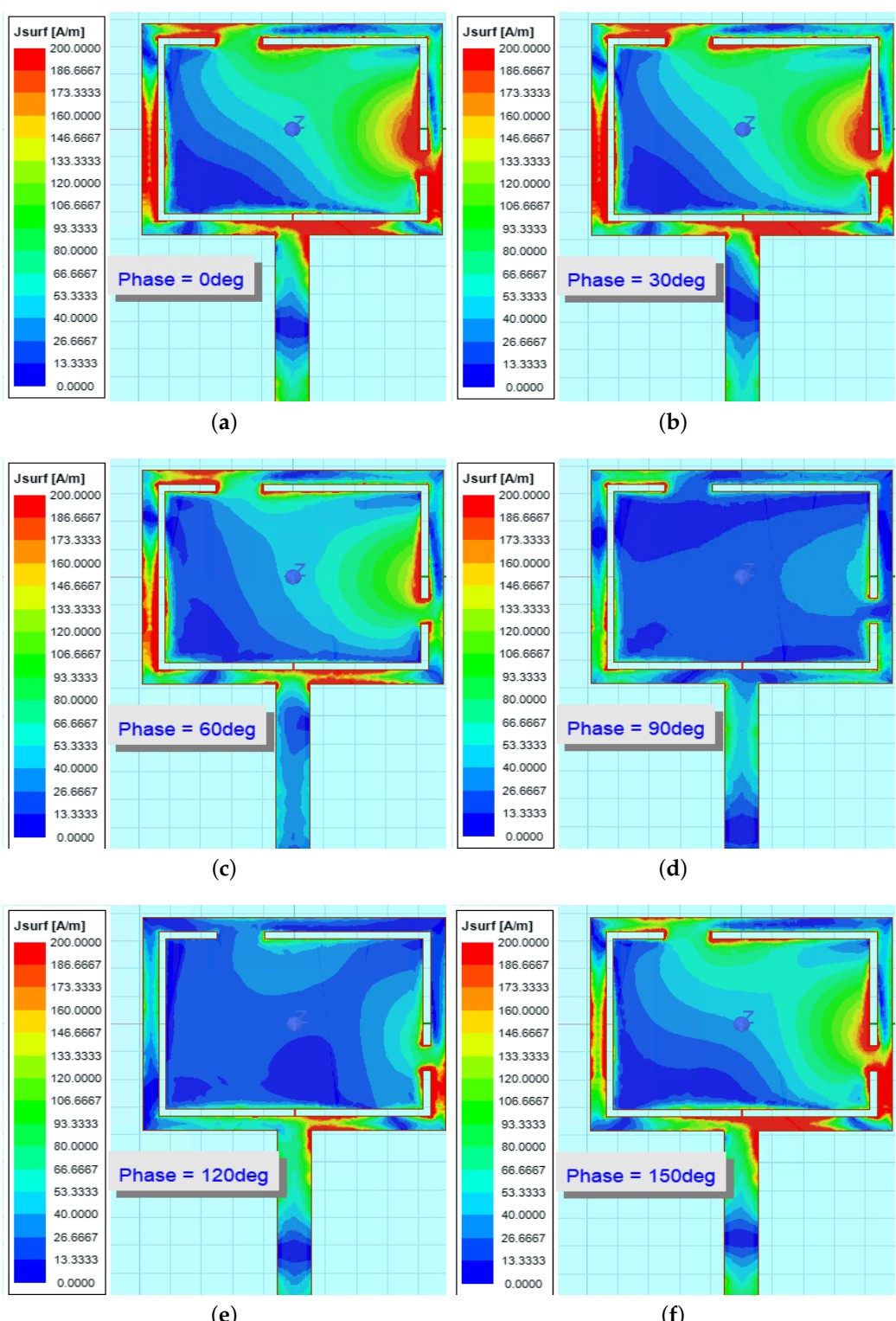

**Figure 17.** Surface current distribution at 6.2 GHz for cylinder radius of 50 mm. (**a**) 0 degrees; (**b**) 30 degrees; (**c**) 60 degrees; (**d**) 90 degrees; (**e**) 120 degrees; and (**f**) 150 degrees.

Figure 19a suggest that the peak gain of the antenna across the frequency sweep range from 4 to 9 GHz. Essentially, the results from the three bending tests have a similar trend, apart from that, the overall peak gain values of the case of 20 mm cylinder radius are slightly lower than those of the other two cases, especially at the frequencies of interest.

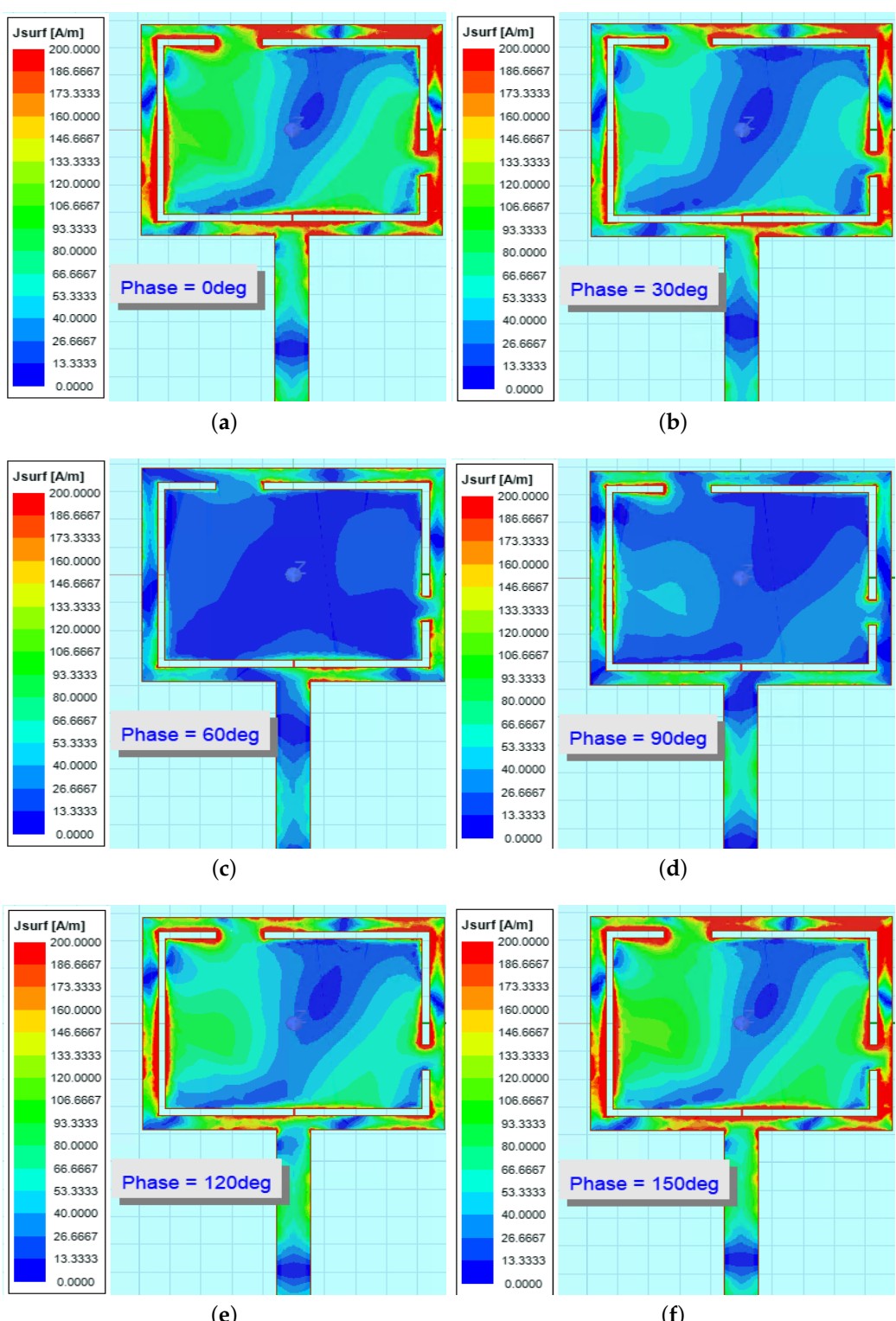

**Figure 18.** Surface current distribution at 8.4 GHz for cylinder radius of 50 mm. (**a**) 0 degrees; (**b**) 30 degrees; (**c**) 60 degrees; (**d**) 90 degrees; (**e**) 120 degrees; and (**f**) 150 degrees.

Figure 19b presents the antenna radiation efficiency for the three bending tests. Taking both conduction loss and dielectric loss into account, the antenna radiation efficiency indicates the portion of input energy that is transmitted by the antenna. From Figure 19b, it is clear that the antenna has high radiation efficiency above 80% at two frequencies of

interest, i.e., 5.8 and 6.2 GHz. At the vicinity of 8 GHz, the radiation efficiency is between 70% and 80%, and the cases of 50 and 80 mm cylinder radius are relatively unstable.

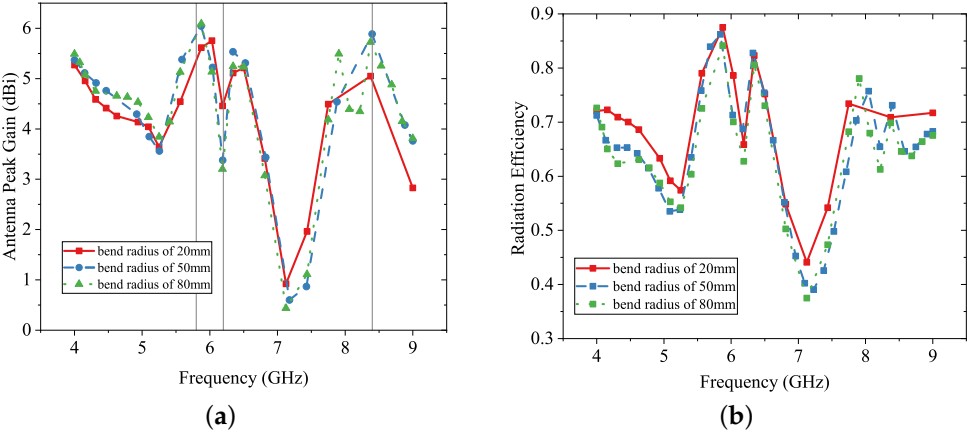

**Figure 19.** Antenna performance for bending tests. (**a**) Antenna peak gain; and (**b**) Antenna radiation efficiency.

Table 1 summarises the main characteristics of the antenna for the conformal bending tests. Three frequencies of interest are individually evaluated as well. These parameters include the maximum directivity $D_{max}$, maximum antenna gain $G_{max}$, radiated power $P_{rad}$ (total input power is set as 1 W), maximum $\theta$ component of E-field $E_{\max}|_\theta$, maximum of total electric field $E_{max}$, maximum radiation intensity $U_{max}$, and radiation efficiency $e_{cd}$. The optimal results among the tests are highlighted in green. Conversely, the minimum values are in red.

**Table 1.** The antenna parameters for conformal tests.

| | $r$ = 20 mm | | | | $r$ = 50 mm | | | $r$ = 80 mm | |
| --- | --- | --- | --- | --- | --- | --- | --- | --- | --- |
| $f$ (GHz) | 5.8 | 6.2 | 8.4 | 5.8 | 6.2 | 8.4 | 5.8 | 6.2 | 8.4 |
| $D_{max}$ (dB) | 3.96 | 3.89 | 4.60 | 4.67 | 2.98 | 5.30 | 4.81 | 3.21 | 5.39 |
| $G_{max}$ (dB) | 3.50 | 2.62 | 3.27 | 4.04 | 2.21 | 3.87 | 4.09 | 2.10 | 3.78 |
| $P_{rad}$ (W) | 0.88 | 0.31 | 0.54 | 0.83 | 0.62 | 0.67 | 0.80 | 0.37 | 0.60 |
| $E_{max}|_\phi$ (V) | −1.12 | 1.40 | 1.26 | 2.30 | −1.85 | 1.36 | 2.30 | 1.22 | −1.82 |
| $E_{max}|_\theta$ (V) | 0.35 | 5.48 | 5.24 | 6.07 | 0.52 | 5.48 | 6.11 | 5.66 | 0.80 |
| $E_{max}$ (V) | 14.43 | 8.52 | 12.20 | 15.20 | 10.52 | 14.55 | 15.20 | 8.42 | 13.92 |
| $U_{max}$ (W/sr) | 0.28 | 0.10 | 0.20 | 0.31 | 0.15 | 0.28 | 0.31 | 0.09 | 0.26 |
| $e_{cd}$ | 0.88 | 0.67 | 0.71 | 0.87 | 0.74 | 0.73 | 0.85 | 0.66 | 0.70 |

Predictably, the scattering of the data highlighted in green in Table 1 shows a noticeable trend to reflect the effect of bending. As the bending cylinder radius increases (i.e., bending angle decreases), the number of optimal results increases as well. However, something unexpected is that, at 6.2 GHz and a bending radius of 80 mm, it also possesses the largest number of minimum results highlighted in red, even though some of the values are very close to each other.

To sum up, all the results of the conformal bending tests suggest that bending surfaces have general negative effects on the antenna performance, and these are proportional to the bending angle. Therefore, a large bending angle or strong surface deformation should be avoided in practical applications. Despite these adverse impacts, the antenna still produces acceptable performance under conformal situations. Furthermore, some results like the antenna parameters at 6.2 GHz for a cylinder radius of 80 mm in Table 1 also suggest that the association between conformal effects and antenna performance is not strictly fixed and predictable. Practical measurements and tests are required for further verification.

## 6. Simulation for Wearable Applications

In the previous section, assessment of the antenna under conformal circumstances with bending curvature was conducted, and the antenna demonstrated acceptable resistance and promising performance. One common application for conformal circumstance is wearable devices for the human body, which requires further evaluation according to various criteria.

In this section, the antenna is evaluated together with part of human body model to simulate its performance as a wearable device with a close distance with human body tissue. Limited by the model mesh complexity and computation time, a male human left hand model from the ANSYS HFSS library is used for simulation. Note that the model applies an approximate average density of 1 g/cm$^3$.

As demonstrated in Figure 20, the hand model is placed under the antenna, which is set bending along the 80 mm cylinder from the previous conformal test. The curvature of the entire antenna has a close match with the hand model. The distance between the antenna and the hand model is set as a variable $d_{ant}$ with values of 0, 10, 20, 30, and 40 mm.

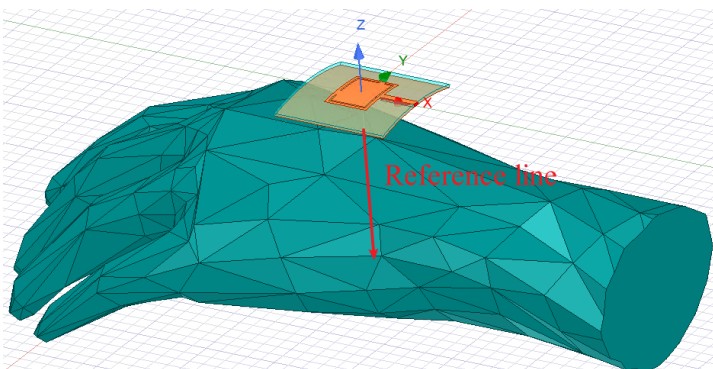

**Figure 20.** Human hand model and the reference line for SAR calculation.

Figure 21 illustrates the 3D plots of E-field strength at three frequencies of interest. It can be seen that the general pattern shapes remain similar to those in the conformal bending tests. The maximum E-field strength is the largest at 8.4 GHz, i.e., 14.9 V/m, followed by 14.8 V/m at 5.8 GHz and 10.6 V/m at 6.2 GHz. Additionally, the patterns also suggest that most of the radiation energy concentrates above the *XoY*-plane, which indicates less absorption if the antenna is attached above a human body.

The 2D SAR plots on the cross-section planes of the hand model are illustrated in Figures 22 and 23 for the *XoZ*-plane and Figure 24 for the *YoZ*-plane. Note that the average SAR calculation is applied, computed by averaging over a volume that surrounds each mesh point in the hand model. To be harmless to the human body, the SAR limit is set as 1.6 W/kg in the USA and 2 W/kg in the EU.

In Figure 22, one can observe clearly that the SARs in most of the cross-section area in the three cases are below 2 W/kg, which means that it is safe for human body tissue. The orange area (1.79–4.06 W/kg) and yellow (0.36–1.79 W/kg) in each diagram are nearly the same size, while the size of the blue region (below 0.0012 W/kg) differs in the three cases. At 8.4 GHz, the plot has the largest blue region, followed by the case at 6.2 and then 5.8 GHz.

As for the *YoZ*-plane SAR plots in Figure 24a–c, similarly, most of the cross-section area is under the safe limit level. A noticeable deviation is that the high SAR regions (red and orange regions) are located differently. For instance, at 5.8 GHz, the left part of the hand model has relatively higher SAR, whereas at 6.2 and 8.4 GHz, it is on the right part. In addition, the size of the blue regions also varies in different cases. For instance, the one at 5.8 GHz is the largest and located in the bottom right region.

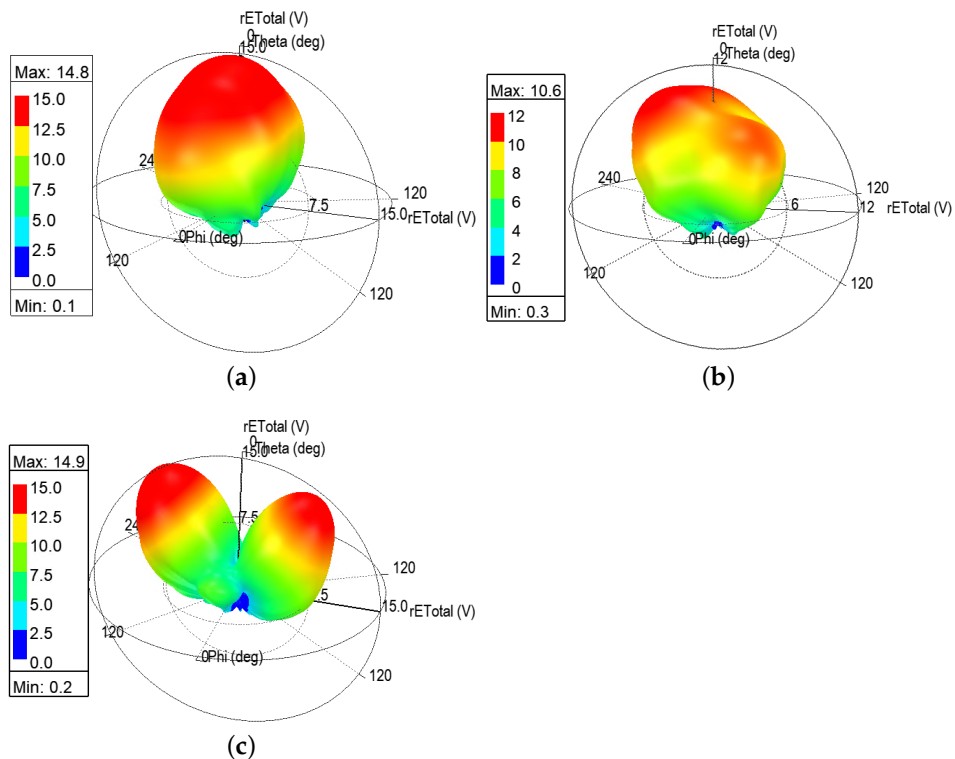

**Figure 21.** E-field 3D plot for $d_{ant}$ = 0 mm. (**a**) E-field 3D plot at 5.8 GHz; (**b**) E-field 3D plot at 6.2 GHz; and (**c**) E-field 3D plot at 8.4 GHz.

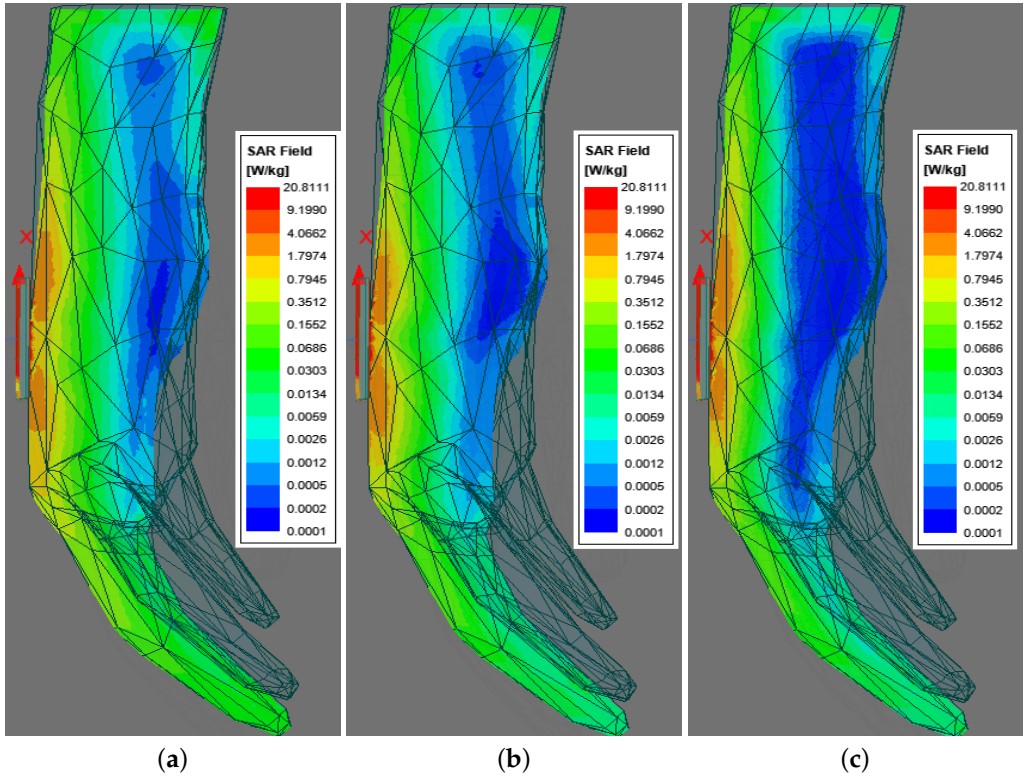

**Figure 22.** SAR$_{avg}$ plot on the *XoZ*-plane when $d_{ant}$ = 0 mm. (**a**) SAR$_{avg}$ at 5.8 GHz; (**b**) SAR$_{avg}$ at 6.2 GHz; and (**c**) SAR$_{avg}$ at 8.4 GHz.

Combining the SAR results on two cross-sections, one can conclude that the overall SAR level inside human the hand model is within the limit, except for a small area on the surface close to the antenna, indicated by the red regions. These above cases represent the situation when the antenna is in close contact with human body, i.e., $d_{ant} = 0$ mm. In practical application, a gap may exist between the antenna and human body; therefore, other cases where $d_{ant} = 10, 20, 30,$ and $40$ mm, are also simulated.

Predictably, as the antenna moves further away from the hand model, the radiation intensity inside the body tissue will decrease. For simplicity of presentation and readability, the 2D SAR plots of the case $d_{ant} = 40$ mm are presented in Figure 23 for the $XoZ$-plane and Figure 24d–f for the $YoZ$-plane.

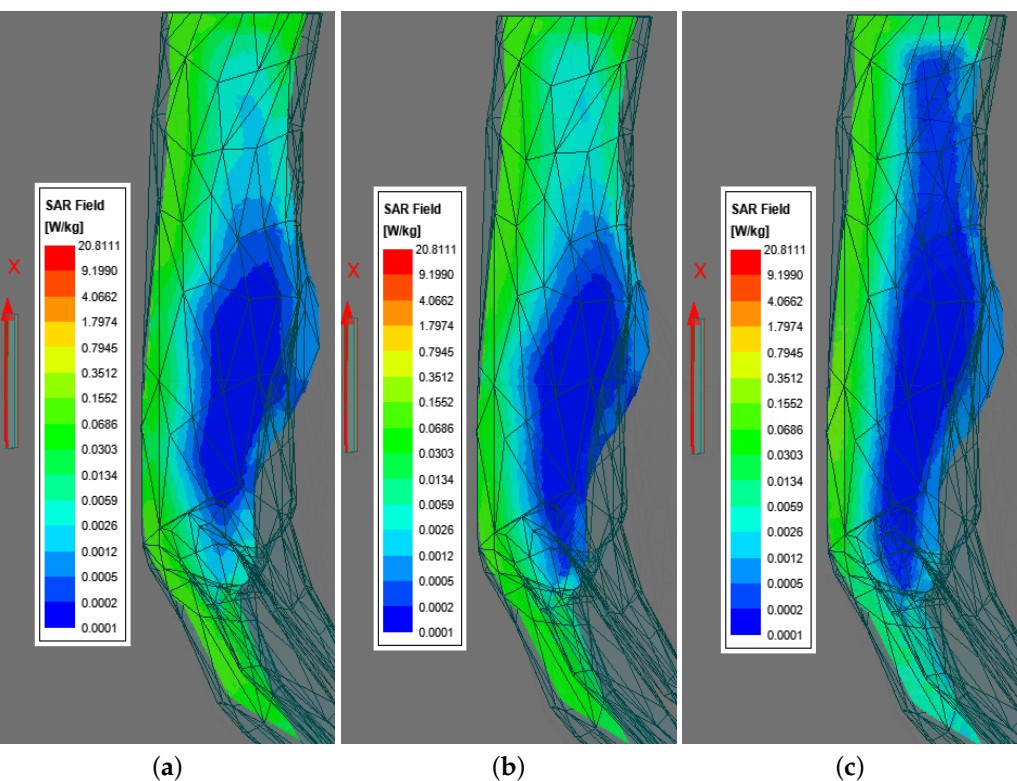

**Figure 23.** SAR$_{avg}$ plot on the $XoZ$-plane when $d_{ant} = 40$ mm. (**a**) SAR$_{avg}$ at 5.8 GHz; (**b**) SAR$_{avg}$ at 6.2 GHz; and (**c**) SAR$_{avg}$ at 8.4 GHz.

For the $XoZ$-plane SAR value, it can be clearly seen in Figure 23 that all the regions for three frequencies of interest are below 0.16 W/kg, which is only 8% of the SAR limit. Additionally, the blue region (below 0.005 W/kg) also takes up to half of the whole cross-section area for at 5.8 and 6.2 GHz, and more than half at 8.4 GHz.

Looking at the plots of the $YoZ$-plane in Figure 24, a similar conclusion can be drawn that the SAR value of entire cross-section area is below 0.16 W/kg, which is far below the SAR limit. What is different from the previous $XoZ$-plane plot is that the blue region at 5.8 GHz is larger than that at 6.2 and 8.4 GHz. This corresponds to the fact that the back lobes of the radiation pattern at 5.8 GHz are rather small compared with those at 6.2 and 8.4 GHz. Consequently, the radiated energy below the antenna is lower as well.

To elaborate the SAR results for different $d_{ant}$, a reference line is created along the $-z$-axis across the hand model as indicated in Figure 20. The SAR values are plotted along the line with respect to $d_{ref}$, the depth from top surface of the hand model along the $-z$-axis.

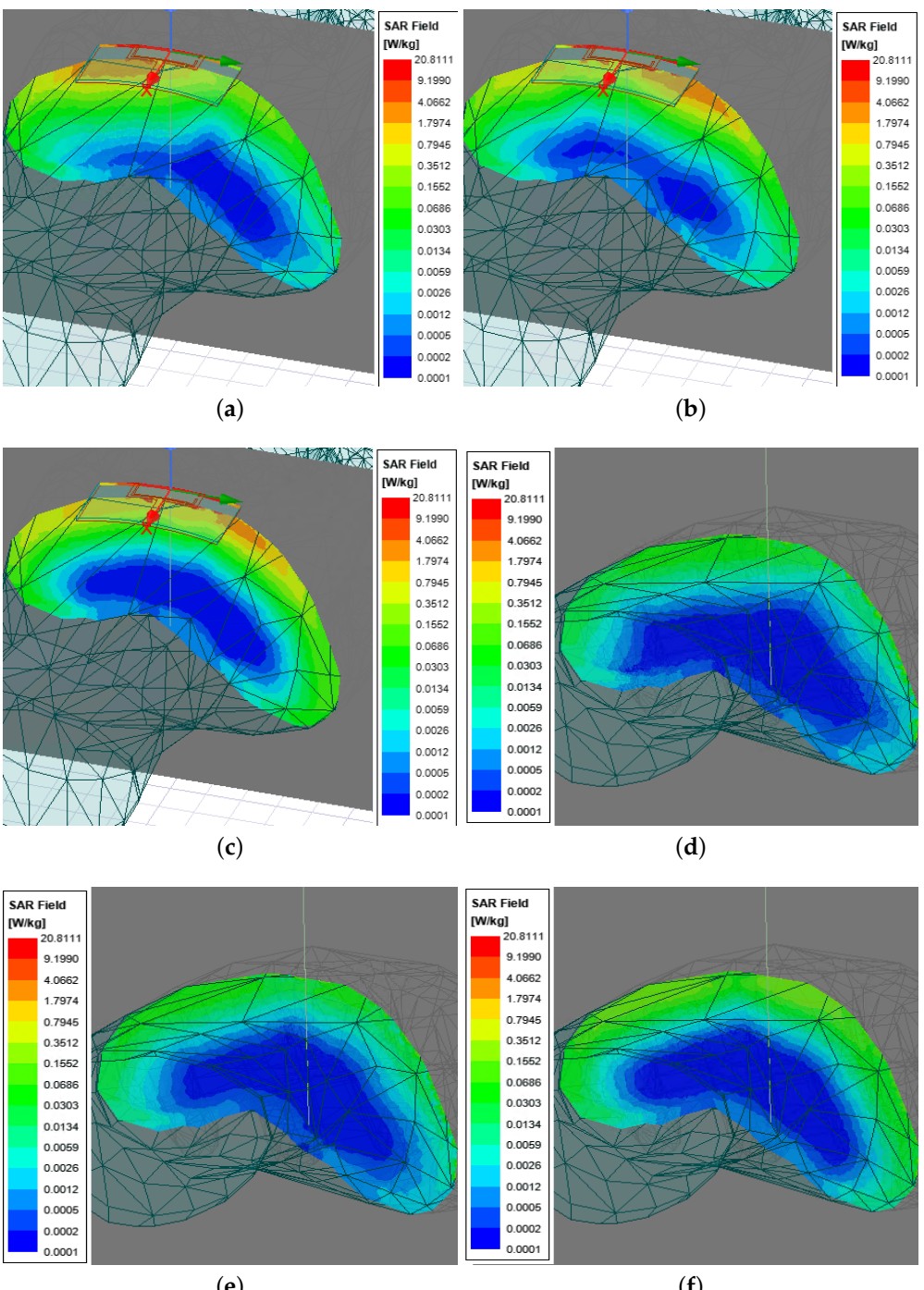

**Figure 24.** $SAR_{avg}$ plot on *YoZ*-plane. (**a**) 5.8 GHz for $d_{ant}$ = 0 mm; (**b**) 6.2 GHz for $d_{ant}$ = 0 mm; (**c**) 8.4 GHz for $d_{ant}$ = 0 mm; (**d**) 5.8 GHz for $d_{ant}$ = 40 mm; (**e**) 6.2 GHz for $d_{ant}$ = 40 mm; and (**f**) 8.4 GHz for $d_{ant}$ = 40 mm.

As can be seen from Figure 25, all the SAR curves approach zero beyond the point $d_{ref}$ = 15 mm, before which lie the variations of each case. In Figure 25a, the SAR curves are above 2 W/kg before the point $d_{ref}$ = 2 mm and up to 40 W/kg, which means that, within 2 mm under the surface, human tissue is absorbing excessive radiated electromagnetic energy, and the amount drops below the threshold for deeper region and reaches zero after the point $d_{ref}$ = 15 mm. Similarly, in Figure 25b, the threshold point is also near $d_{ref}$ = 2 mm. Moreover, a steep decline of SAR happens at this point.

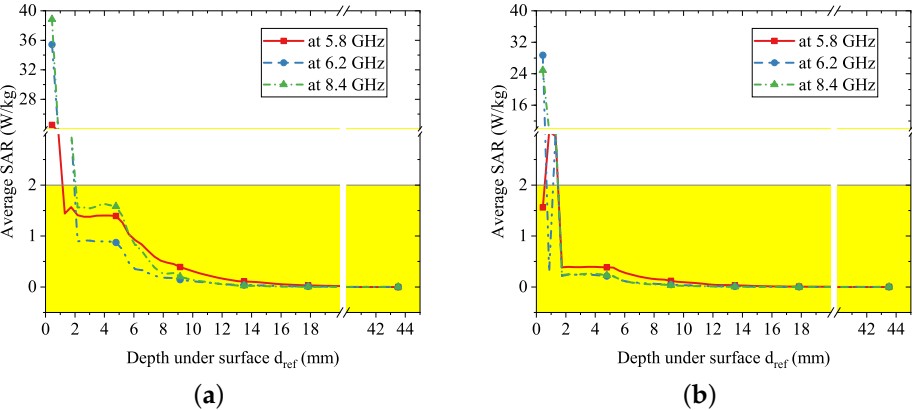

**Figure 25.** Average SAR vs. $d_{ref}$ along the reference line. (**a**) SAR on reference line for $d_{ant}$ = 0 mm; and (**b**) SAR on reference line for $d_{ant}$ = 10 mm.

However, the results for the other three cases where $d_{ant}$ is 20, 30, and 40 mm, the SAR value in the entire region is below 2 W/kg. The only difference is the decreasing tendency before the threshold $d_{ref}$ = 15 mm. The trends in Figure 26a,c are similar, where the overall SAR value of the 8.4 GHz curve is the highest and has a steep decline, followed by the 5.8 GHz one and then the 6.2 GHz one, both of which have a rather smooth decline. Surprisingly, for the case of $d_{ant}$ = 30 mm as shown in Figure 26b, the curve of 5.8 GHz is above the one of 8.4 GHz. All three curves have a steep decline.

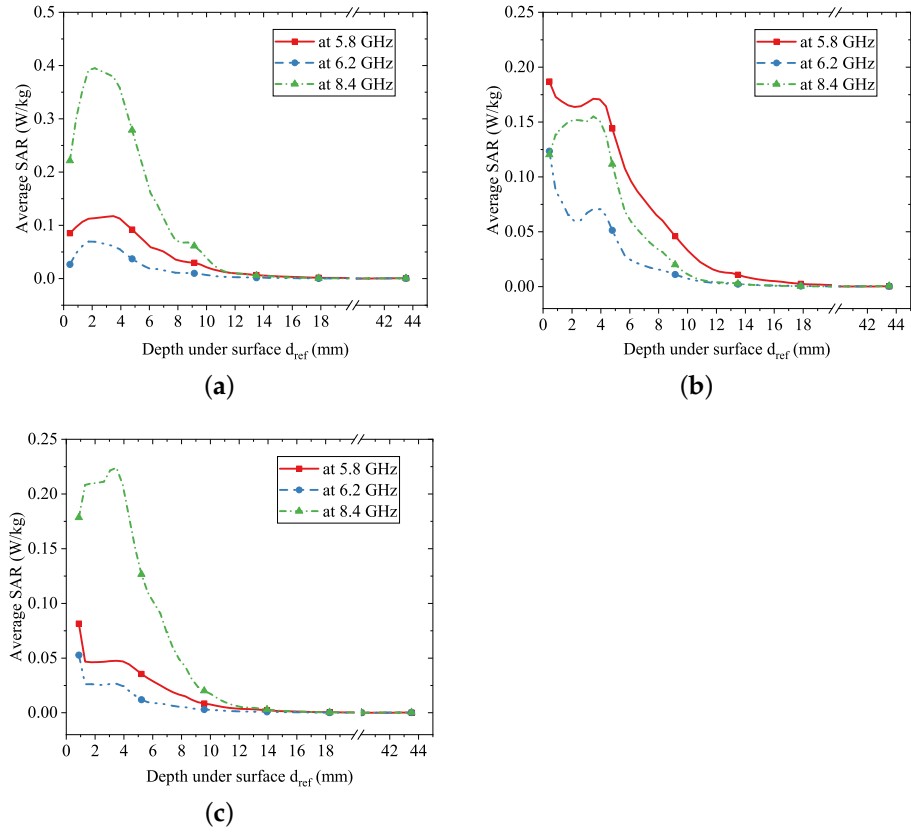

**Figure 26.** Average SAR vs. $d_{ref}$ along the reference line. (**a**) SAR on reference line for $d_{ant}$ = 20 mm; (**b**) SAR on reference line for $d_{ant}$ = 30 mm; and (**c**) SAR on reference line for $d_{ant}$ = 40 mm.

The above results demonstrate promising performance of the proposed antenna for on-body wearable applications. Embedding the antenna with other devices can be easily achieved due to its compact size and flexible structure. Moreover, since we proposed the antenna with a flexible PET film as a substrate, which has a low price (approximately 1.4 USD per kilogram with a thickness from 0.1 to 3 mm), the major implementation cost is the fabrication cost. The antenna can be fabricated with economic printing techniques, such as screen printing and inkjet printing.

## 7. Conclusions

We presented a triband slot antenna design operating at 5.8, 6.2, and 8.4 GHz. Design theory and parametric study were given for the design procedure. The antenna demonstrated promising performance in simulation results. As the antenna was proposed with a flexible substrate for future conformal situations and potential wearable applications, successive exploration and simulation were conducted.

The conformal tests with a cylinder bending surface indicate that the antenna performance underwent some fluctuations at different levels of conformal surfaces; however, the overall results were acceptable. Furthermore, the simulations integrated with a human body model suggest promising performance for wearable applications with certain conditions, such as the distance between the antenna and human body.

Due to the facility restrictions and time limit, the proposed triband slot patch antenna was not measured for practical conformal circumstances and wearable application with a real human body—only simulated results were provided. The fabrication and measurements will be part of our future work. Additionally, our future work will also include the evaluation of the antenna operating with other devices in a complete wireless communication system, such as the interference between other sensors, terminal devices, and the environment.

**Author Contributions:** Conceptualization, E.L. and X.J.L.; methodology, E.L.; software, E.L.; validation, E.L., X.J.L. and B.-C.S.; formal analysis, E.L., X.J.L.; investigation, E.L., B.-C.S.; resources, B.-C.S.; data curation, E.L.; writing—original draft preparation, E.L.; writing—review and editing, X.J.L., B.-C.S.; visualization, X.J.L., B.-C.S.; supervision, X.J.L., B.-C.S.; project administration, X.J.L., B.-C.S. All authors have read and agreed to the published version of the manuscript.

**Funding:** This research received no external funding.

**Conflicts of Interest:** The authors declare no conflict of interest.

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
