# Peer review of "A Triband Slot Patch Antenna for Conformal and Wearable Applications"

_electronics, doi:10.3390/electronics10243155_

Round 1

Reviewer 1 Report

The paper is very interesting, well organized, and the scientific content is current and in line with the state of the art. In the paper they wrote about the importance in the field of IoT, but the applicability of your antenna in this field is redundant, you can demonstrate how the antenna can be applied in a real scenario of IoT or other technologies that can bring benefits to your solution. A new antenna is designed, but interference, signal-to-noise ratio, among other parameters for evaluating telecommunications systems, are not analyzed. I think they should improve this part.

minor issues:

  • At the end of the introduction, write the organization of the paper's chapters;
  • The number of references is short, improve with citations from research papers and commercial solutions;

Reviewer 2 Report

Authors have presented research on “A Triband Slot Patch Antenna for Conformal and Wearable Applications”. Following comments will be helpful to further improve the manuscript and add clarity.

  • Overall, the manuscript is presented well, in detail with bending and SAR analysis.
  • Add motivation for the proposed design.
  • The proposed work is for 5G applications; however, the targeted frequencies (5.8 GHz, 6.2 GHz, and 8.4 GHz) are not 5G commercial frequencies. How this work is 5G?
  • It is not clear that which results are measured, and which are simulated. Please clearly mention this in captions of figures, simulated/measured for each result.
  • Measured results are missing.
  • Scale values in some figures are not clearly visible due to smaller font size, i.e., in 3D patterns, current distributions and SAR. Scale font size needs to be bigger for better visibility.

Reviewer 3 Report

The authors focus their study on designing a triband patch antenna with multiple slots in order to support wearable applications. The authors exploit the Internet of Things applications and the proposed antenna operates at several frequencies and it has been tested and simulated in order to quantify its efficiency.

The manuscript is overall well written and easy to follow and the authors have well thought out their contributions. The design of the proposed antenna has been clearly described and the authors have provided all the details in order to enable the reader to understand the main contributions of the authors.

The authors should consider the following suggestions provided by the reviewer in order to improve the scientific depth of their manuscript, as well as  the quality of its presentation.

Initially, the authors should better motivate the need of the design of this antenna in the Internet of Things applications.

There are several theoretical models that have been introduced in the literature in order to collect data stemming from Internet of Things devices, such as Tsiropoulou, E. E., et al. "Interest, energy and physical-aware coalition formation and resource allocation in smart IoT applications." 2017 51st Annual Conference on Information Sciences and Systems. IEEE, 2017, and the authors should better clarify how the proposed design will support existing applications.

Furthermore, the authors should include an additional subsection in their manuscript describing the implementation cost in order this antenna to be implemented in a realistic set up. This is critical in order for the proposed implementation to be adopted in a realistic environment.

Finally, the overall manuscript should be checked for typos, syntax, and grammar in order to improve the quality of its presentation.

Round 2

Reviewer 2 Report

Authors have addressed the comments and manuscript have now been improved. However, few things still need to be clearly mentioned in the manuscript as suggested earlier. Paper may be accepted after below changes.

It must clearly be stated that results are simulated, and measured results are not presented.

Change the following captions as:

Figure 9 (d): “Simulated return loss and gain of proposed antenna.”

Figure 10 (a): “Simulated radiation pattern at 5.8 GHz.”

Figure 10 (c): “Simulated radiation pattern at 6.2 GHz.”

Figure 10 (e): “Simulated radiation pattern at 8.4 GHz.”

Figure 10: “Simulated antenna performance at three bands.”

Figure 11: “Simulated antenna return loss for different bending radii.”

Figure 15: “Simulated E-plane and H-plane for conformal tests.”

Reviewer 3 Report

The authors have addressed in detail the reviewers’ comments and the quality of presentation, as well as the scientific depth of the paper have been substantially improved. This reviewer has no further concerns regarding the publication of this paper.
